# Comparative Analysis on the Evolution of Flowering Genes in Sugar Pathway in Brassicaceae

**DOI:** 10.3390/genes13101749

**Published:** 2022-09-28

**Authors:** Yingjie Zhang, Qianbin Zhu, Hao Ai, Tingting Feng, Xianzhong Huang

**Affiliations:** 1College of Life Sciences, Shihezi University, Shihezi 832003, China; 2Center for Crop Biotechnology, College of Agriculture, Anhui Science and Technology University, Chuzhou 233100, China

**Keywords:** ephemeral plant, *Arabidopsis pumila*, flowering, sugar metabolism, purifying selection

## Abstract

Sugar plays an important role in regulating the flowering of plants. However, studies of genes related to flowering regulation by the sugar pathway of Brassicaceae plants are scarce. In this study, we performed a comprehensive comparative genomics analysis of the flowering genes in the sugar pathway from seven members of the Brassicaceae, including: *Arabidopsis thaliana*, *Arabidopsis lyrata*, *Astelia pumila*, *Camelina sativa*, *Brassica napus*, *Brassica oleracea*, and *Brassica rapa*. We identified 105 flowering genes in the sugar pathway of these plants, and they were categorized into nine groups. Protein domain analysis demonstrated that the *IDD8* showed striking structural variations in different Brassicaceae species. Selection pressure analysis revealed that sugar pathway genes related to flowering were subjected to strong purifying selection. Collinearity analysis showed that the identified flowering genes expanded to varying degrees, but *SUS4* was absent from the genomes of *Astelia pumila*, *Camelina sativa*, *Brassica napus*, *Brassica oleracea*, and *Brassica rapa*. Tissue-specific expression of *ApADG* indicated functional differentiation. To sum up, genome-wide identification revealed the expansion, contraction, and diversity of flowering genes in the sugar pathway during Brassicaceae evolution. This study lays a foundation for further study on the evolutionary characteristics and potential biological functions of flowering genes in the sugar pathway of Brassicaceae.

## 1. Introduction

Polyploidy events have occurred frequently during plant speciation [1]. Dicotyledons experienced the (γ) event [2,3], and monocotyledons underwent the Tau (τ) event [4,5]. Polyploidy events can enhance the adaptability of plants to tolerate extreme environmental changes [6,7]. As the largest flowering plant family on Earth, the family Brassicaceae has undergone whole-genome duplications (At-α) [8,9], while *Brassica* L. crops have even undergone an additional genome-wide triploid duplication (Br-β) [10,11]. Comparative genomic analysis has shown that *C. sativa* is an allohexaploid plant, derived from the hybridization of three ancestral *Arabidopsis* species, and it has undergone an additional genome-wide triploid duplication [12,13].

Flowering is an important part of the life cycle of angiosperms, affecting the transmission of genetic information, the size of population and the genetic diversity of species [14]. The flowering process is mainly divided into three stages: flower induction, flowering transition, and flower organ differentiation. Flowering is comprehensively affected by both plant internal signals and environmental factors. In *Arabidopsis*, Bouché et al. [15] systematically identified eight pathways that promote flowering: aging, ambient temperature, circadian clock, hormones, autonomy, photoperiod, vernalization, and sugar pathways. The aging pathway is mainly regulated by the microRNAs (miRNAs) *miR156-SPL* and *miR172-AP2* [16]. The ambient temperature pathway is mainly affected by the interaction between two spliceosome proteins of *FLOWERING LOCUS M* and *SHORT VEGETATIVE PHASE* [17]. The response of flowering in plants to light signals requires the interaction between the photoperiod and circadian clock pathways [18]. Studies on the hormone pathways have mainly focused on the gibberellin (GA) signaling pathway. GA not only promotes the development of plant organs, but also induces flowering. However, the effect of GA on different plants is not consistent [19,20]. An autonomous pathway mainly affects flowering through inhibiting the expression of *FLOWERING LOCUS C* [21]. The research on the vernalization pathway has primarily focused on *FLOWERING LOCUS C* and *FRIGIDA* [22].

Research on the sugar pathway and flowering has largely focused on the synthesis and transport of sugars, but the contribution of sugars to flowering regulation is still poorly understood. In the sugar pathway, the genes *ADP GLUCOSE PYROPHOSPHORYLASE 1* (*ADG1*), *PHOSPHOGLUCOSE ISOMERASE 1* (*PGI1*), and *PHOSPHOGLUCOMUTASE* (*PGM1*) promote starch synthesis under active photosynthesis [23]. *SUCROSE SYNTHASE 4* (*SUS4*) can also promote starch synthesis [24] and *INDETERMINATE DOMAIN 8* (*IDD8*) regulates the expression of *SUS4* [25]. *SNF1 KINASE HOMOLOG 10* (*SNRK1.1*) inactivates *IDD8* through protein phosphorylation, thereby delaying flowering [26]. Carbohydrates respond to *HEXOKINASE 1* (*HXK1*) through the SYMPORTER 9 (SUC9) protein as a signaling molecule in the form of sucrose [27], thereby repressing *miR156* expression and promoting the flowering transition [28], or after conversion of sucrose to trehalose-6-phosphate (T6P), with the involvement of *TREHALOSE-6-PHOSPHATE SYNTHASE 1* (*TPS1*) [29].

Although the effect of sugar on flowering time has been studied many times, how sugar regulates flowering time is unclear. Either sugar promotes flowering or represses flowering depends on sucrose concentration [30], localization of invertases [31], and T6P levels [32,33]. In addition, sugar stimulates the expression levels of genes involved in regulating the circadian clock, such as *GIGANTEA*, *TIMING OF CAB EXPRESSION 1*, and *CIRCADIAN CLOCK ASSOCIATED 1* [34,35]. So, sugar affects the flowering transition in multiple ways. However, the molecular mechanism by which sugars regulate flowering and fruiting in members of the Brassicaceae is still unclear.

To date, the genomes of more than 1000 species, including the model plant *A. thaliana*, have been sequenced and assembled [36,37,38], including those of other members of the Brassicaceae, such as *A. lyrata* [39], *C. sativa* [12], *B. napus* [40], *B. oleracea* [41], and *B. rapa* [42]. *A. pumila*, an ephemeral plant of the Brassicaceae and a distant relative of *A. thaliana* [43,44], is mainly distributed in the deserts of the Junggar Basin in northern Xinjiang in China [45,46]. To analyze its molecular mechanism for adaptation to the arid desert environment, we have performed a transcriptomic study of leaf tissues of *A. pumila* in response to continuous salt stress [46]. In recent years, although there have been reports of gene members in the sugar pathway that regulate flowering, there has been little research conducted on the identification and comparative genomic analysis of gene members of this pathway regulating the transition to flowering in the Brassicaceae, especially in the recently sequenced species. In this study, we performed genome-wide identification of flowering genes in the sugar pathway from seven Brassicaceae species: *A. thaliana*, *A. lyrata*, *A. pumila*, *C. sativa*, *B. napus*, *B. oleracea*, and *B. rapa*. Bioinformatics analyses, such as protein domain, gene structure, and comparative evolution, were carried out to reveal their characteristics during evolution. This study provides a theoretical basis for further study of interspecific phylogeny and variation and the mining of flowering genes in the sugar pathway of Brassicaceae.

## 2. Materials and Methods

### 2.1. Plant Materials

Seeds of *A. pumila* were collected in May 2012 from the southern margin (44°20′ N and 87°46′ E) of the Gurbantunggut Desert in Xinjiang, China, as described by Yang et al. [47]. *A. pumila* seeds were surface sterilized and planted as described by Jin et al. [48]. Seven-day-old seedlings were transplanted into pots containing peat soil and vermiculite (1:1) and placed in an illuminated incubator at 22 °C under long-day (LD) conditions (16 h light/8 h dark) with 300 μmol m^−2^ s^−1^ light intensity. *A. pumila* tissue samples were collected at different stages, from embryos (at 2 d after sowing), young roots (8 d), mature roots (30 d), rosette leaves (30 d), cauline leaves (60 d), buds (60 d), flowers (65 d), siliques (70 d), and seeds (70 d), for gene expression analysis. Immediately after collection, the samples were snap-frozen with liquid nitrogen and then stored in a −80 °C freezer.

### 2.2. Identification and Naming of Gene Members

We referred to FLOR-ID to identify flowering gene members in the sugar pathway of *A. thaliana* (http://www.phytosystems.ulg.ac.be/florid/, accessed on 10 November 2021) [15], and genomic data were downloaded from the TAIR database (http://www.arabidopsis.org, accessed on 10 November 2021) (Appendix A). *A. lyrata* and *C. sativa* genomic data were downloaded from Ensemble Plant (http://plants.ensembl.org/index.html, accessed on 10 November 2021). *B. napus*, *B. oleracea,* and *B. rapa* genomic data were downloaded from BRAD (http://brassicadb.cn/#/, accessed on 10 November 2021). The coding sequence (CDS) and protein sequencer files of *A. pumila* were provided by our groups. Algorithm-based BLASTP was performed using the amino acid sequences of *A. thaliana* SUS and SUC proteins as queries in the protein databases of *A. lyrata, A. pumila*, *C. sativa*, *B. napus*, *B. oleracea*, and *B. rapa*, with an *E* < 1 × 10^−5^ and other parameters as default values. The candidate protein sequences were compared with Pfam (http://pfam.xfam.org/, accessed on 27 April 2022) database using HMMER (http://www.hmmer.org/, accessed on 27 April 2022). Identification of the orthologous genes between *A. thaliana* and the other six species used the Ortho Venn2 web tool (http://www.bioinfogenome.net/OrthoVenn/, accessed on 27 April 2022) for analysis. The flowering gene members in the sugar pathway, the *SUS* gene family members, and the *SUC* gene family members of six plant species were identified based on the clustering of homologous genes with those of *A. thaliana*. Flowering gene members in the sugar pathway were also named according to the homologous genes with *A. thaliana*. Names of the *SUS* and *SUC* gene family members are listed in Appendix A. The physicochemical properties of the protein encoded were analyzed by the ProtParam tool of ExPASy (https://web.expasy.org/protparam/, accessed on 28 April 2022). The subcellular localization of flowering proteins was predicted by Cell-PLOC 2.0 (http://www.csbio.sjtu.edu.cn/bioinf/Cell-PLoc-2/, accessed on 28 April 2022).

### 2.3. Phylogenetic Tree Construction, Gene Structure, and Protein Motif Analysis

The ClusterW program [49] was used to perform multiple sequence alignments between the flowering protein sequences in the sugar pathway of *A. thaliana*, *A. lyrata*, *A. pumila*, *C*. *sativa*, *B. napus*, *B. oleracea*, and *B. rapa* with default parameters. MEGAX [50] was used to construct Neighbor-Joining phylogenetic trees and to explore the evolutionary relationship between sugar pathway flowering genes among different species. The bootstraps test was carried out with 1000 iterations. The phylogenetic trees were visualized using the Interactive Tree of Life (iTOL) program (https://itol.embl.de/, accessed on 29 April 2022). Based on the genome gff3 files of *A. thaliana*, *A. lyrata*, *A. pumila*, *C. sativa*, *B. napus*, *B. oleracea*, and *B. rapa* using the Gene Structure Display Server (GSDS) (http://gsds.cbi.pku.edu.cn/, accessed on 29 April 2022) visualization server, the distribution of gene exons-introns was analyzed. The conserved motifs of protein were identified using the MEME website (http://meme-suite.org/tools/meme, accessed on 29 April 2022). The motif length range was set to 10–60 amino acid residues, and the maximum number of motif discoveries was set to 10, while other parameters were set to default values.

### 2.4. Select Pressure and Duplication Type Analysis

MCscanX [51] was used to calculate the collinearity of homologous genes between species, with a threshold of *E* < 10^−5^, while its downstream program duplicate_gene_classifier was used to analyze the duplication type of genes. Easy_KaKs program was used to calculate the synonymous substitution ratio (*K*s), non-synonymous substitution ratio (*K*a), and *K*a/*K*s ratio; the evolution mode was judged according to the size of the *K*a/*K*s ratio [52]. Tbtools [53] was used to visualize the collinearity relationship among gene members of *A. thaliana* and the other six species.

### 2.5. Gene Expression Analysis and Quantitative Real-Time-PCR (qPCR) Validation

*A. pumila* transcriptome sequencing data were downloaded from the BioProject database (https://www.ncbi.nlm.nih.gov/bioproject, accessed on 8 June 2021) (Appendix A), and the transcriptome accession number of the nine tissues was PRJNA721579. Sequencing samples were collected from the following tissues of *A. pumila*: embryos (2 d after sowing), young roots (8 d), mature roots (30 d), rosette leaves (30 d), cauline leaves (60 d), buds (60 d), flowers (65 d), siliques (70 d), and seeds (70 d).

The fastp tools were used to filter and compare sequencing data [54], and the comparison was achieved using the Bowtie2 tool [55]; parameters are set to default values. The results were standardized using the fragments per kilobase of transcript per million mapped reads (FPKM) of a gene. After the FPKM value was converted by Log2(FPKM + 1), a heatmap was created using the R package ‘Pheatmap’ (https://CRAN.R-project.org/package=pheatttmap, accessed on 1 May 2022), and the expression of flowering genes in the sugar pathway among different tissue was analyzed.

qPCR primers were designed according to the gene CDS sequence, and the *ApGADPH* gene was used as the internal reference gene (Appendix A) [48]. RNAprep Pure Plant Kit (Tiangen Biotech, Beijing, China) was used to extract RNA. M-MLV RTase cDNA Synthesis Kit (Takara, Dalian, China) was used to reverse transcription. RNA extraction and reverse transcription followed the methods of Huang et al. [45]. qPCR was carried out with the SYBR Green PCR Master Mix system (Takara) on an ABI ViiA7 real-time fluorescence quantitative PCR machine (Life Technologies, New York, NY, USA). The procedure of qPCR and the analysis method of experimental data were carried out by referring to the method of Jin et al. [48]. Three biological replicates were performed, with RNA isolated independently, and each qPCR reaction had three replicates.

### 2.6. Protein Interaction Network Prediction

The Ortho Venn2 tool (http://www.bioinfogenome.net/OrthoVenn/, accessed on 27 April 2022) was used to identify the orthologous genes between *A. pumila* and *A. thaliana*. Then, interaction networks of *A. pumila* flowering genes in the sugar pathway were identified based on the orthologous genes between *A. pumila* and *A. thaliana*, using STRING (https://string-db.org/, accessed on 3 May 2022) databases, and the predicted protein-protein interaction network was displayed Cytoscape software [56].

## 3. Result

### 3.1. Homology-Based Identification of Flowering Genes in the Sugar Pathway in Seven Species of the Brassicaceae and their Evolution

Genome-wide analyses identified 105 genes from seven species of the Brassicaceae associated with flowering in the sugar pathway, based on their homology with the nine known genes of this pathway in *A. thaliana*. These included 8 in *A. lyrata*, 16 in *A. pumila*, 22 in *C. sativa*, 24 in *B. napus*, 13 in *B. oleracea*, and 13 in *B. rapa* (Appendix A), the number of homologs of each *A. thaliana* gene varying between the species (Figure 1). The highest numbers of homologs were for genes *SNRK1.1* and *ADG1*, whereas *SUS4* and *SUC9* exhibited the fewest homologs (Appendix A). Phylogenetic analysis of the *SUS* and *SUC* gene families in the Brassicaceae showed that *SUS4* was absent from all species except *A. thaliana* and *A. lyrata*, while only *A. thaliana* contained the *SUC9* gene (Appendix A). Phylogenetic analysis also revealed that, with the exception of *ADG1*, *SNRK1.1,* and *PGI1*, the sugar pathway gene members formed two well-separated clusters, the first composed of sequences from *A. thaliana*, *A. lyrata*, *A. pumila*, and *C. sativa* and the second of sequences from *B. napus*, *B. oleracea*, and *B. rapa* (Figure 1 and Figure 2A). The physicochemical properties and subcellular localization details of their corresponding proteins are summarized in Appendix A.

### 3.2. Gene Structure and Protein Conserved Protein Motifs

Gene structure comparisons showed that the flowering genes in the sugar pathway were relatively highly conserved but that there were substantial differences in the number of exons in different genes (Figure 2B), ranging from three in *IDD8* to 22 in *PGM1*. The number of exons in *PGM1*, *SUC9*, *IDD8*, and *SUS4* genes in the seven species studied was conserved among the species, whereas the number of exons in other genes showed species-related differences. For example, *AtHXK1*, *AlHXK1*, *ApHXK1*, and *CsHXK1* all contain seven exons, while *BnHXK1*, *BoHXK1*, and *BrHXK1* contain eight or nine exons. Based on the differences in exon number, the seven species of the Brassicaceae could be broadly clustered into two groups, one consisting of *A. thaliana*, *A. lyrata*, *A. pumila*, and *C. sativa*, and the other consisting of *B. napus*, *B. oleracea*, and *B. rapa*.

Comparative analysis of motif structure showed that most proteins of each species are highly conserved (Figure 2C). The distributions and numbers of motifs in ADG1 and HXK1 of the seven species are completely consistent, but differences were observed in other proteins. Relative to the majority consensus protein structures in the seven species, CsPGM1-1 has no motif 4; BoPGI1-1 has no motif 2; BoPGI1-2 lacks both motif 9 and motif 10; BrTPS1-1 lacks motif 1; ApSNRK1.1 shows a loss of motif 10 and an altered position for motif 1. The IDD8 homologs displayed the most striking differences between the species, involving loss or gain of motifs with substantial differences in inter-motif distances.

### 3.3. Collinear Relationship of Flowering Gene Members

To investigate the contraction and expansion of flowering gene members in the sugar pathway during evolution, the relationship of orthologous genes between *A. thaliana* and six species of the Brassicaceae was explored using MCScanX software [51]. The gene members with collinear relationships in each plant species were only distributed on some chromosomes (Figure 3). *A. thaliana* has 8, 14, 20, 20, 11, and 12 collinear genes with *A. lyrata*, *A. pumila*, *C. sativa*, *B. napus*, *B. oleracea*, and *B. rapa*, respectively (Appendix A). Except for *SUC9*, the eight flowering genes of *A. thaliana* show collinearity with *A. lyrata* (Figure 3A). In *A. pumila*, except for the two missing genes, namely *AtSUS4* and *AtSUC9*, each of the other seven genes has two collinear gene pairs with *A. thaliana*. In *C. sativa*, the *IDD8* gene has two collinear gene pairs with *A. thaliana*, whereas each of the other six flowering genes of the sugar pathway has three collinear gene pairs with *A. thaliana*. The comparison shows that *B. oleracea* and *B. rapa* showed the same number of homologs of their *A. thaliana* counterparts (minus *SUS4* and *SUC9*), three for *SNRK1.1*, two for *TPS1* and *HXK1*, and one for each of the others (Figure 3B). However, in *PGI1*, only one collinear gene pair occurred between *A. thaliana* and *B. oleracea*, with two collinear gene pairs between *A. thaliana* and *B. rapa* (Figure 3B). Compared with *A. thaliana*, there were four collinear gene pairs of *TPS1*, *SNRK1.1*, *PGI1*, and *HXK1*, two of *IDD8*, and one each of *ADG1* and *PGM1* in *B. napus*. Overall, the number of flowering genes in *B. napus* was approximately three times that of *A. thaliana* and twice that of *B. oleracea* or *B. rapa* (Figure 3B).

### 3.4. Selection Pressure

The *K*a/*K*s ratios of the homologous genes between *A. thaliana* and each of the other six plant species are less than 1 (Figure 4), indicating that these homologous genes are subject to strong purifying selection. The average *K*a/*K*s value of the *TPS1* collinear gene pairs was the smallest, indicating that they suffer from relatively strong purifying selection, whereas the average *K*a/*K*s value of the *IDD8* gene pair was the highest, indicating a weaker purifying selection (Appendix A). However, only *A. thaliana* and *A. lyrata* had *K*a/*K*s values for *SUS4* genes. The *K*a/*K*s values for *TPS1*, *HXK1,* and *IDD8* genes pairs were similar between the species, while the *K*a/*K*s values for *SNRK1.1*, *PGI1*, *ADG1*, and *PGM1* gene pairs exhibited clearly larger variations among the species, suggesting that these genes may have undergone functional divergence during species evolution.

### 3.5. Gene Duplication Type

The duplication types of the 105 genes in the seven species of Brassicaceae were determined. The results showed that the duplication types of these genes could be divided into three types: whole-genome duplication/segmental duplication (WGD/S), dispersed duplication, and tandem duplication (Table 1). WGD/S events in the seven species, namely *A. thaliana*, *A. lyrata*, *A. pumila*, *C. sativa*, *B. napus*, *B. oleracea*, and *B. rapa*, were 6 (66.7%), 4 (50%), 18 (100%), 24 (96%), 9 (85.7%), 10 (69.2%), and 24 (79.6%), respectively. To summarize, with regard to flowering genes of the sugar pathway, the main gene duplication events in *A. lyrata* were dispersed duplication and WGD/S, whereas the main process underlying gene duplications in the other six species was WGD/S.

### 3.6. Tissue Expression Profiles of the Flowering Genes in the Sugar Pathway in A. pumila

Using the RNA-Seq data from nine tissues of *A. pumila*, the tissue expression heatmap of 16 *A. pumila* genes was constructed (Figure 5A). The results showed that most genes showed a trend of high expression in all tissues at the early stages of growth, whereas the number of genes with high expression at the late stage of growth was relatively small. Moreover, the expression patterns of most gene duplicates were the same. For example, *ApADG1-3*/*4* and *ApPGM1-1*/*2* were both highly expressed in the rosette and cauline leaves, *ApPGI1-1*/*2* and *ApIDD8-1*/*2* were highly expressed in young and mature roots, and *ApSNRK1.1-1*/*2* was highly expressed in both flowers and siliques (Figure 5). However, some duplicate gene pairs were differentially expressed in the same tissue. For example, *ApADG1-1* was highly expressed in siliques, whereas *ApADG1-2* expression was highest in embryos, and *ApAHX1.2* was more highly expressed in the rosette and cauline leaves than *ApHXK1-1*.

qPCR was further used to validate the expression determined from transcriptomics of *ApADG1-1*/*2*/*4* in different tissues (Figure 5B). The results from the two methods consistently confirmed the accuracy of the transcriptomic data.

### 3.7. Protein Interaction Network

In order to better understand the biological function of the sugar pathway proteins, we next predicted the interaction network of the proteins associated with flowering regulation in the sugar pathway. *Arabidopsis* PGM2 and PGM3 proteins can interconvert glucose-6-phosphate (G6P) and glucose-1-phosphate (G1P) (Figure 6). The *Arabidopsis pgm2 pgm3* double mutant has been reported to markedly damage the function of male and female gametophytes [57]. ENOLASE 2 (ENO2) maintains normal growth, development, and successful reproduction in plants and is also able to help plants respond to abiotic stresses such as drought and cold [58,59,60]. ENOLASE 1 (ENO1) plays a crucial role in the development of gametophytes and sporophytes [61]. TRIOSE PHOSPHATE ISOMERASE (TPI) promotes the transition of plants from heterotrophic growth to autotrophic growth [62]. The *Arabidopsis snf4* mutation disrupts the interaction of SNF4 with SNRK1 and thus affects the role of SNRK1 in the T6P-mediated signaling pathway [63].

## 4. Discussion

In this study, an evolutionary analysis of those genes of the sugar pathway that were also involved in the regulation of flowering in seven species in the Brassicaceae revealed that a total of nine genes were identified: *TPS1*, *SNRK1.1*, *SUS4*, *PGI1*, *HXK1*, *SUC9*, *IDD8*, *ADG1*, and *PGM1* (Figure 1). Analysis showed a large number of genes in *SNRK1.1* and *ADG1* (Appendix A), indicating that these contributed largely to the gene expansion in the sugar pathway observed in the Brassicaceae species studied, other than *A. thaliana* and *A. lyrata*. In contrast, *SUS4* and *SUC9* were found only in *A. thaliana* and *A. lyrata* (Figure 1). However, *SUS1* and *SUS4* in *A. thaliana* both belong to the *SUS I* subfamily and have very similar gene structures and amino acid sequences [64]. Therefore, we speculate that the loss of function caused by the deletion of *SUS4* in these five plant species may be complemented by the increased expression of *SUS1*, a hypothesis that needs to be further tested. In the *SUC* gene family, the six Brassicaceae species share common ancestors in *AtSUC6*, *AtSUC7*, *AtSUC8*, and *AtSUC9* of the *SUT1* subfamily [65]. In *A. thaliana*, both *AtSUC8* and *AtSUC9* were expressed in flowers, and the corresponding loss-of-function mutants *Atsuc8* and *Atsuc9* did not differ from the wild type in terms of flowering time and rosette size [66]. It is suggested that there may be genes that functionally substitute for or exhibit functional redundancy with *AtSUC9* in *A. lyrata*, *A. pumila*, *C. sativa*, *B. napus*, *B. oleracea*, and *B. rapa*, but with different evolutionary branches from that of *A. thaliana* (Appendix A).

The number of exons in each sugar pathway gene was largely similar in each of the plant species, indicating that these genes were relatively conserved during their evolution/expansion in the Brassicaceae. However, the organization and loss/gain of IDD8 motifs was seen to vary greatly between the different species, suggesting that IDD8 may have undergone neofunctionalization during evolution (Figure 2). The sugar pathway genes of the seven species were shown to have been subjected to strong purifying selection, which further supports the finding that these genes were highly conserved during evolution (Figure 4, Appendix A).

WGD events have been observed to occur frequently during plant evolution [67,68]. In the current study, we found that the number of members of the sugar pathway genes had expanded in some Brassicaceae species. We identified eight, 16, 13, 13, 24, and 22 flowering genes in the sugar pathway from the genomes of *A. lyrata*, *A. pumila*, *B. oleracea*, *B. rapa*, *B.napus*, and *C. sativa*, respectively. After removing the missing genes *SUS4* and *SUC9* from consideration, the number of the flowering genes in the sugar pathway in these six spp. is 1.0, 2.3, 1.9, 1.9, 3.4, and 3.1 times that of *A. thaliana*, respectively. The results showed that, with the occurrence of WGD events, the number of flowering genes in the sugar pathway had also increased. In *A. lyrata*, the main duplication type of the flowering genes was dispersed duplication, which is different from that in *A. thaliana*, *A. pumila, C. sativa*, *B. napus*, *B. oleracea*, and *B. rapa*, with WGD/S as the main duplications (Table 1), indicating that WGD/S played a leading role in the expansion of flowering genes in the sugar pathway of the Brassicaceae.

The genome size of *A. lyrata* is about twice that of *A. thaliana* [36,39], but the number of genes identified in these two species is similar. The number of genes identified in *C. sativa* is about three times that of *A. thaliana*, indicating a triploidy event relative to *A. thaliana*. This finding is consistent with the research results of Kagale et al. [12]. However, the number of genes in the sugar pathway of *B. napus* is inconsistent with the fact that this species is an allotetraploid [69], and the numbers of genes in *B. oleracea* and *B. rapa* are also inconsistent with the current evidence that they have experienced ancient triploidy events [10,11]. The results of collinearity analysis showed that the number of *Bo**SNRK1.1* and *BrSNRK1.1* genes was three times that of *AtSNRK1.1*, whereas other genes showed no three-fold greater number than that in *A. thaliana*, indicating that flowering genes in the sugar pathway of *B. oleracea* and *B. rapa* may have been lost after the triploidy event (Figure 3). This result indicated that *SNRK1.1* plays an important role in the sugar metabolism of *B. oleracea* and *B. rapa*, which is in agreement with the previous findings of *A. thaliana* [70]. There is no six-fold relationship in the number of any of the flowering regulatory genes between *B. napus* and *A. thaliana*, which also indicates that some of the duplicated flowering genes in the sugar pathway of *B. napus* were lost after the duplication event (Figure 3). *B. napus* is formed by the natural crossing of *B. rapa* (AA) and *B. oleracea* (CC) and subsequent tetraploidy [69]. Furthermore, 11 and 9 sugar pathway genes were located in the AA and the CC subgenome, respectively, suggesting that they played a similarly vital role in both ancestral species.

The diversity of gene expression patterns in different tissues suggests that flowering genes in the sugar pathway are involved in many aspects of the growth and development of *A. pumila* in addition to regulating flowering transition (Figure 5). For example, *ApPGI1-1*/*2* and *ApIDD8-1*/*2* were specifically expressed in roots during vegetative growth, a finding that is consistent with the trend of high expression of *AtPGI* and *ApIDD8* in roots in Arabidopsis DNA microarray data (http://www.genevestigator.com, accessed on 30 May 2022). The result indicated that the functions of *ApPGI1-1/2* and *ApIDD8-1/2* are similar to those of *AtPGI1* and *AtIDDD8* [23,26]. In addition, some orthologous genes showed different expression profiles in the same tissues (Figure 5), indicating that, although there are certain functional similarities between the orthologous genes, functional differentiation has occurred during evolution. For example, *ApADG1-1* is highly expressed in siliques, while *ApADG1-2* is highly expressed in embryos. Similarly, *ApADG1-4* is highly expressed in the rosette and cauline leaves, compared with *ApADG-1/2* (Figure 5). In *A. thaliana*, *ADG1* promotes the transition from a vegetative-to-reproductive phase of development [71], indicating that *ApADG1-1*/*2* underwent functional differentiation during evolution. *ApADG1-2* may promote seed germination, whereas *ApADG1-1* appears to promote the maturation of siliques, a finding that needs further experimental verification.

Sugar distribution is the basis of plant growth and development and also plays an important role in flowering transition [72]. The analysis of protein interactions revealed that, in *A. pumila*, the proteins interacting with flowering regulatory proteins are mainly functional in the metabolism of carbohydrates and signaling, suggesting that the distribution of sugars plays key roles in plant growth and development, flowering transition, and tolerance of abiotic stresses (Figure 6). Four of these proteins, namely ApPGI1, ApPGM1, ApHXK1, and ApADG1, were located at the core of the protein interaction network, suggesting that they may be involved in the transcriptional regulation of more genes and hence in more metabolic regulation in plants (Figure 6).

In summary, we comprehensively identified the flowering gene members of the sugar pathway from seven Brassicaceae species and analyzed their structural characteristics and evolutionary characteristics. The results provide clues to further exploring the molecular mechanism(s) underlying the regulation of flowering by sugar metabolism.

## Figures and Tables

**Figure 1 genes-13-01749-f001:**
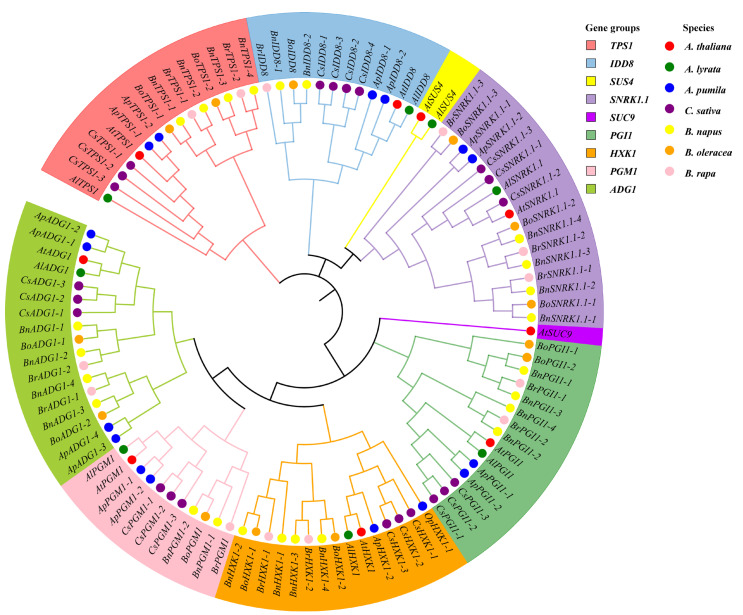
Neighbor-joining phylogenetic tree of the flowering genes in the sugar pathway from seven members of Brassicaceae.

**Figure 2 genes-13-01749-f002:**
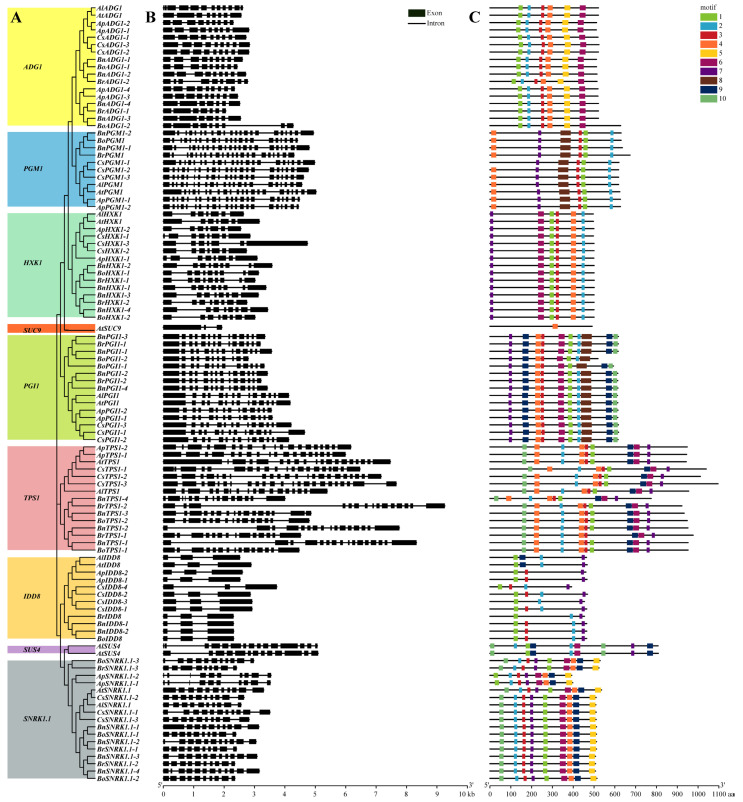
Phylogenetic tree, exon-intron structures, and motif distributions of the flowering genes in the sugar pathway in *A. thaliana*, *A. lyrata*, *A. pumila*, *C. sativa*, *B. napus*, *B. oleracea*, and *B. rapa*. (**A**) Phylogenetic tree; (**B**) exon-intron distribution; (**C**) distribution characteristics of the conserved motifs; aa, amino acid.

**Figure 3 genes-13-01749-f003:**
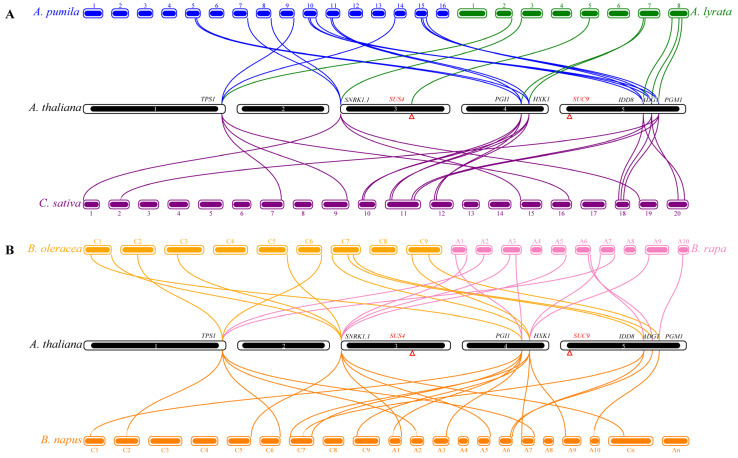
Syntenic relationships of sugar pathway genes associated with flowering in *A. thaliana*, *A. lyrata*, *A. pumila*, *C. sativa*, *B. napus*, *B. oleracea,* and *B. rapa*. Different lines represent collinear gene pairs between different species. Blocks of different colors represent the chromosomes of different plant species. Red triangles in chromosomes 4 and 5 of *A. thaliana* represent the locations of *SUS4* and *SUC9* genes. (**A**) Syntenic relationships of sugar pathway genes associated with flowering in *A. thaliana*, *A. lyrata*, *A. pumila*, and *C. sativa*; (**B**) Syntenic relationships of sugar pathway genes associated with flowering in *A. thaliana*, *B. napus*, *B. oleracea*, and *B. rapa*.

**Figure 4 genes-13-01749-f004:**
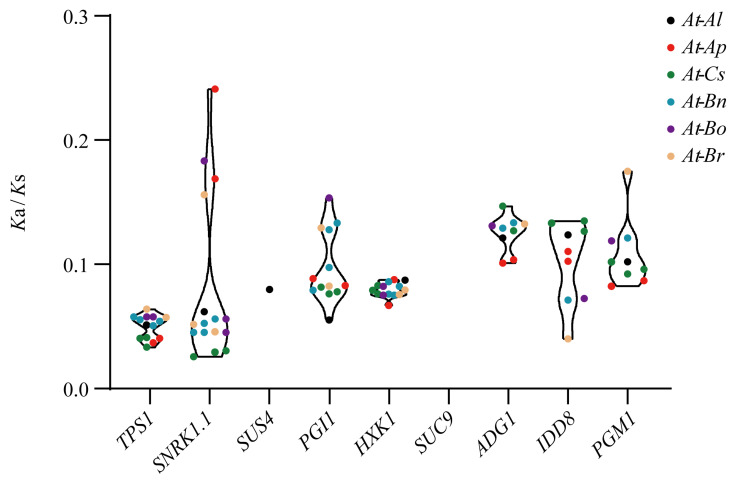
*K*a/*K*s values exhibited by flowering genes in the sugar pathway among species. *At*, *A. thaliana*; *Al*, *A. lyrata*; *Ap*, *A. pumila*; *Cs*, *C. sativa*; *Bn*, *B. napus*; *Bo*, *B. oleracea*; *Br*, *B. rapa*.

**Figure 5 genes-13-01749-f005:**
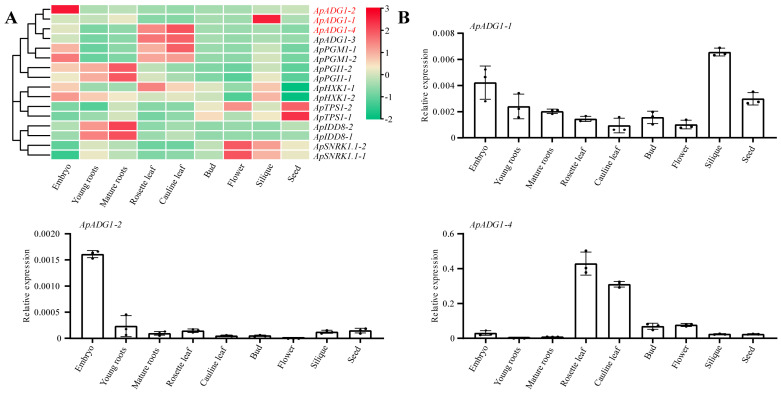
Tissue expression profiles of the flowering genes in the sugar pathway in *A. pumila*. (**A**) Tissue expression heatmap of flowering genes; (**B**) qPCR analyses of the relative gene expression levels of *ApADG1* homologs in different tissues. The data points represented mean ± SD.

**Figure 6 genes-13-01749-f006:**
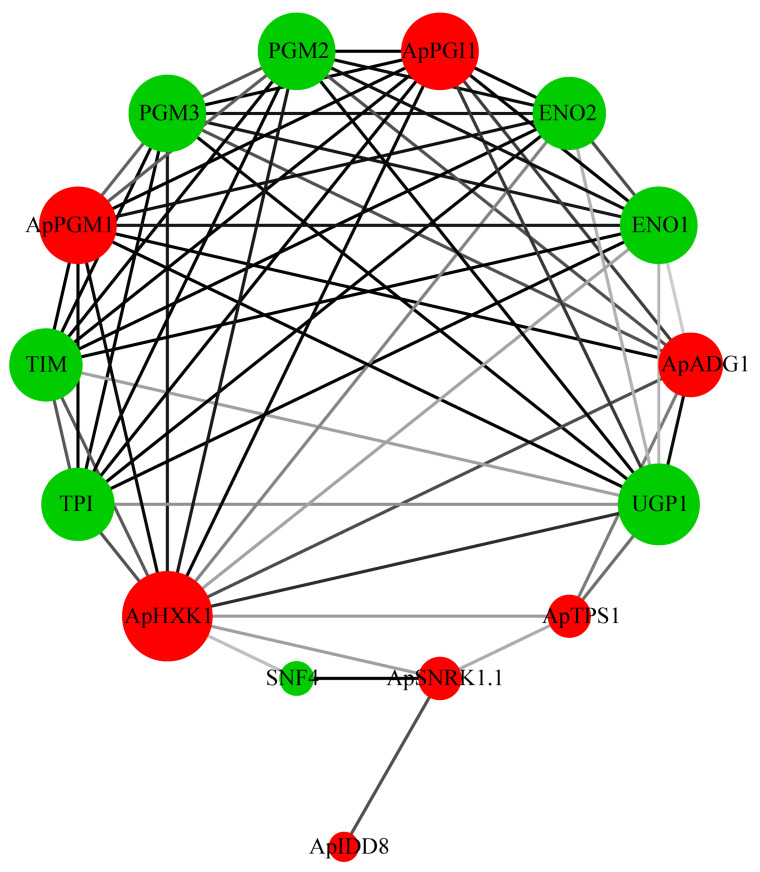
Interaction network diagram of flowering regulatory proteins in the sugar pathway. The darkness of the lines is proportional to the combined_score (between 0 and 1, where the higher the value, the greater the reliability of the predicted interaction between the two proteins). The node size increases with the number of interacting proteins. Red circles are proteins that are members of the *A. pumila* sugar pathway; green circles are interacting with *A. thaliana* proteins.

**Table 1 genes-13-01749-t001:** Types of duplication of the flowering genes in the sugar pathway in the seven species of Brassicaceae.

Species	No.of Sugar Pathway Genes	No. of Sugar Pathway Genes from Different Duplication Type Origins (Percentage)		
		Dispersed	Tandem	WGD/S
*A. thaliana*	9	3 (33.3)	0	6 (66.7)
*A. lyrata*	8	4 (50)	0	4 (50)
*A. pumila*	18	0	0	18 (100)
*C. sativa*	25	0	1 (4)	24 (96)
*B. napus*	28	4 (14.3)	0	24 (85.7)
*B. oleracea*	13	3 (23.1)	1 (7.7)	9 (69.2)
*B. rapa*	13	3 (23.1)	0	10 (76.9)

## Data Availability

Flowering gene members in the sugar pathway of *A. thaliana* were downloaded from FLOR-ID (http://www.phytosystems.ulg.ac.be/florid/, accessed on 10 November 2021). *A. thaliana* genomic data were downloaded from the TAIR database (http://www.arabidopsis.org, accessed on 10 November 2021). *A. lyrata* and *C. sativa* genomic data were downloaded from Ensemble Plant (http://plants.ensembl.org/index.html, accessed on 10 November 2021). *B. napus*, *B. oleracea,* and *B. rapa* genomic data were downloaded from BRAD (http://brassicadb.cn/#/, accessed on 10 November 2021). *A. pumila* transcriptome sequencing data were downloaded from the BioProject database (https://www.ncbi.nlm.nih.gov/bioproject, accessed on 8 June 2021).

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
