# Peer review of "Comparative Analysis on the Evolution of Flowering Genes in Sugar Pathway in Brassicaceae"

_genes, 2022, doi:10.3390/genes13101749_

Round 1

Reviewer 1 Report

The research (A comparative analysis of the evolution of the flowering genes in the sugar pathway of Brassicaceae) employed the molecular mechanisms underlying the relationship between sugar metabolism and flowering regulation from seven Brassicaceae plants, including: Arabidopsis thaliana, A. lyrata, A. pumila, Camelina sativa, Brassica napus, B. oleracea, and B. rapa, and confirmed the molecular mechanism underlying the regulation of flowering by sugar metabolism.

 The manuscript idea is good but the manuscript needs major revisions:

1.     Abstract is poor and needs re-writing with clear results, not general information.

2.     The authors must mention the A. pumila seeds source.

3.     The authors must mention the seeds growing conditions before referencing Jin et al. [41].

4.     Figures have very bad resolutions and are not accepted for publishing.

5.     Discussion is poor and needs fortification with new references.

6.     Correct references 24, 28, and 61, the year didn’t mention two times in the reference.

7.     The manuscript needs English language and grammar revision.

Author Response

Dear editors and reviewers,

Thank you for your reviews on our manuscript, “A comparative analysis of the evolution of the flowering genes in the sugar pathway of Brassicaceae”. These comments greatly helped us improve our manuscript and provided important guidance for future research. The revised portions were used the “Track Changes” in the revised manuscript.

According to your reviews, we have carefully revised our manuscript one-by-one. First, we improved the abstract section to emphasize the important findings. We added the seeds details for A. pumila. For the picture quality, we have designed each figure based on the requirements of Genes magazine and hope they meet with the demands. We also revised improved the discussion section. The corresponding references have been cited and edited. For the language and grammar quality, we have requested the International Science Editing office to polish our manuscript. At last, we modified the formats of reference.

The following is the response of the point-to-point review:

1, Abstract is poor and needs re-writing with clear results, not general information.

Response: Thanks. According to your reviews, we have revised the part of Abstract carefully to concrete study results under the 200 words limit.

2, The authors must mention the A. pumila seeds source.

Response: Thank you for your instructive suggestions. In the revised version, we have added a description for A. pumila seeds source and cited the related reference (Yang, L.F.; Jin Y.H., Huang W, Sun Q, Liu F., Huang X.Z. Full-length transcriptome sequences of ephemeral plant Arabidopsis pumila provides insight into gene expression dynamics during continuous salt stress. BMC Genomics. 2018, 19, 717.)

3, The authors must mention the seeds growing conditions before referencing Jin et al. [41].

Response: Thanks. We have added a description for the growth conditions of A. pumila seeds in “Material and Methods”.

4, Figures have very bad resolutions and are not accepted for publishing.

Response: Thank you very much! The results presented in figures are poor readable. This is due to the compression of the PDF format. We have uploaded a higher resolution version for each figure. In fact, we have designed each figure based on the requirements of Gene magazine.

5, Discussion is poor and needs fortification with new references.

Response: Thanks. In the revised manuscript, we carefully revised discussion and added some new references to enrich our discussion.

6, Correct references 24, 28, and 61, the year didn’t mention two times in the reference.

Response: Thanks. We have corrected the above references according to the requirements of the journal.

7, The manuscript needs English language and grammar revision.

Response: Thanks. We have requested the International Science Editing office to polish our manuscript. We believe that this revision improves the language quality and meets the publish demands.

Reviewer 2 Report

I had a great opportunity to review manuscript entitled: 'A comparative analysis of the evolution of the flowering genes in the sugar pathway of Brassicaceae' which is considered for publication  in Genes journal. The goal of this manuscript is to present and characteristics of flowering genes in the sugar pathway of seven Brassicaceae spp.: Arabidopsis thaliana, A.lyrata, A. pumila, Camelina sativa, Brassica napus, B. oleracea, and B. rapa. Authors presented gene identification, phylogenetic evolution, and functional gene differentiation. All parts of the manuscript is interesting and clearly summarize new data valuable for the research community. The manuscript is well written and represents an important study that can offer insights for future research.

GENERAL COMMENTS:
TITLE
The paper title is well stated, it is informative and concise.

ABSTRACT, INTRODUCTION
Abstract and Introduction were well written.

MATERIAL AND METHODS
Material and research methods are presented appropriately and clearly. Experimental setup and the description in the methods section are well structured, and the statistical analysis is done alright.

RESULTS
The results obtained in this study are interesting. Results presented correctly. The results presented in all figure are poor readable- This is probably due to the compression of the PDF format.

DISCUSSION
In general, the discussion of results is correct and sufficient.

LITERATURE
The items of literature included in the paper are rather sufficient and adequate to the subject of the paper.

The manuscript has high formal standard.
Please verify the correctness of the literature and make a linguistic correction of the text by native speaker.

Author Response

Dear editors and reviewers,

Thank you for your reviews on our manuscript, “A comparative analysis of the evolution of the flowering genes in the sugar pathway of Brassicaceae”. These comments greatly helped us improve our manuscript and provided important guidance for future research. The revised portions were used the “Track Changes” in the revised manuscript.

According to your reviews, we have carefully revised our manuscript one-by-one. For the picture quality, we have designed each figure based on the requirements of Genes magazine and hope they meet with the demands. The corresponding references have been cited and edited. For the language and grammar quality, we have requested the International Science Editing office to polish our manuscript.

The following is the response of the point-to-point review:

1, The results presented in all figure are poor readable.

Response: Thank you very much! I think that the poor quality of figures may due to the compression of the PDF format. In fact, we designed each figure based on the demands for pictures in Gene magazine. We have uploaded a higher resolution version.

2, Please verify the correctness of the literature and make a linguistic correction of the text by native speaker.

Response: Thanks. We have requested the International Science Editing office to polish our manuscript. We believe that this revision improves the language quality and meets the publish demands.

Reviewer 3 Report

In this manuscript, the authors studied the molecular mechanisms underlying the relationship between sugar metabolism and flowering regulation, and performed comprehensive comparative genomics analysis of the flowering genes in the sugar pathway from seven Brassicaceae plants, including: Arabidopsis thaliana, A. lyrata, A. pumila, Camelina sativa, Brassica napus, B. oleracea, and B. rapa. About, 105 flowering genes were identified in the sugar pathway in these plants, and they were categorized into nine groups. Protein domain analysis demonstrated that the IDD8 gene showed striking structural variations in different Brassicaceae species. Selection pressure analysis revealed that sugar pathway genes related to flowering were subjected to strong purifying selection. Collinearity analysis showed that the identified flowering genes expanded in varying degrees, but SUS4 was lost in the genomes of A. pumila, C. sativa, B. napus, B. oleracea, and B. rapa.

- The manuscript is well-written and the design is good. The manuscript yielded some important data. I suggest the following revisions;

- The introduction lacks enough information on the literature and hypothesis of this research. Please expand that.

- The methods are summarized. More details on the whole methodologies need to be included.

- Figures 1, 2, 4, 5 and 6 are of low quality and invisible. Please replace those figures with high quality ones. 

- Figure 3 comes after Figure 4. Re-order the figures and its relevant information

- Please add some more paragraphs to the Discussion section discussing the results and the mechanisms assayed.

- References should be updated accordingly.

Author Response

Dear editors and reviewers,

Thank you for your reviews on our manuscript, “A comparative analysis of the evolution of the flowering genes in the sugar pathway of Brassicaceae”. These comments greatly helped us improve our manuscript and provided important guidance for future research. The revised portions were used the “Track Changes” in the revised manuscript.

According to your reviews, we have carefully revised our manuscript one-by-one. First, we improved the introduction section to expand research. We added more details for methods. For the picture quality, we have designed each figure based on the requirements of Genes magazine and hope they meet with the demands. In addition, we changed the order of Figure 3 and Figure 4. We also revised improved the discussion section. The corresponding references have been cited and edited. For the language and grammar quality, we have requested the International Science Editing office to polish our manuscript. Finally, we update the reference.

The following is the response of the point-to-point review:

1, The introduction lacks enough information on the literature and hypothesis of this research. Please expand that.

Response: Thanks. According to your suggestion, we revised the introduction, added some results and cited the related references to expand this research.

2, The methods are summarized. More details on the whole methodologies need to be included.

Response: Thanks. According to you reviews, we have made a careful revision for the part of materials and methods. In the revised manuscript, we carefully described the materials and methods used in this study and tried to explain them clearly and concretely, such as seed source, seeds growth condition, qPCR reagents, and so on.

3, Figures 1, 2, 4, 5 and 6 are of low quality and invisible. Please replace those figures with high quality ones.

Response: Thank you very much! The results presented in figures are poor readable. This is due to the compression of the PDF format. We have uploaded a higher resolution version. In addition, in figure 1, we switched to a lighter background color so that the words can be more clearly rendered.

4, Figure 3 comes after Figure 4. Re-order the figures and its relevant information.

Response: Thank you for your constructive comments. In the revised manuscript, we rearranged the order of Figures 3 and 4 and made related changes to the text.

5, Please add some more paragraphs to the Discussion section discussing the results and the mechanisms assayed.

Response: Thanks. In the revised manuscript, we added some new references to enrich our discussion, and add one paragraphs to discussing the results.

6, References should be updated accordingly.

Response: Thanks. We have updated the references, according to all referees’ suggestions. Also, the formats for all references were amended in the revised manuscript.

Round 2

Reviewer 1 Report

The authors didn't correct the manuscript as required.

In the abstract part the authors need to clarify the conclusion of the study, what are the biological functions of flowering genes in the sugar pathway? 

Author Response

Dear editors and reviewers,

Thank you for your second reviews on our manuscript, “A comparative analysis of the evolution of the flowering genes in the sugar pathway of the Brassicaceae”. We checked all references and confirmed that they were relevant to the content of our manuscript. The revised portions for this manuscript were marked up using the “Track Changes” function of Word.

According to your reviews, we have carefully revised our manuscript in the abstract. The following is the response of the reviews:

Comment: In the abstract part the authors need to clarify the conclusion of the study, what are the biological functions of flowering genes in the sugar pathway?

Response: Thank you for your suggestions. In the revised version of the abstract, we tried to revise each sentence to emphasize our findings and clarify the conclusion of the study. The biological functions of flowering genes in the sugar pathway need to be further studied. We believe that we can understand their functions and molecular mechanisms for controlling flowering through constant efforts in the future.

Reviewer 3 Report

The revised version is greatly improved as per my suggested comments 

Author Response

Dear editors and reviewers,

Thank you for your second reviews on our manuscript, “A comparative analysis of the evolution of the flowering genes in the sugar pathway of the Brassicaceae”. We checked all references and confirmed that they were relevant to the content of our manuscript. The revised portions for this manuscript were marked up using the “Track Changes” function of Word.

The following is the response of the reviews:

Comment: The revised version is greatly improved as per my suggested comments.

Response: Thank you very much for your positive comments! Your affirmation is our motivation to move forward.